# Logic Unveils Truth, While Disguise Obscures It: Transition Logic Augmented Response Selection for Multi-Turn Dialogue

**Tingchen Fu**[♠◇☾†], **Xueliang Zhao**[♡†], **Lemao Liu**[◇], **Rui Yan**[♠☆*]

[♠]Gaoling School of Artificial Intelligence, Renmin University of China
[♡]Peking University [◇]Tencent AI Lab
[☆]Beijing Key Laboratory of Big Data Management and Analysis Methods
[☾]Engineering Research Center of Next-Generation Intelligent Search and Recommendation,
Ministry of Education
lucas.futingchen@gmail.com    zhaoxlpku@gmail.com

## Abstract

Multi-turn response selection aims to retrieve a response for a dialogue context from a candidate pool and negative sampling is the key to its retrieval performance. However, previous methods of negative samples tend to yield false negatives due to the one-to-many property in open-domain dialogue, which is detrimental to the optimization process. To deal with the problem, we propose a sequential variational ladder auto-encoder to capture the diverse one-to-many transition pattern of multiple characteristics in open-domain dialogue. The learned transition logic thus assists in identifying potential positives in disguise. Meanwhile, we propose a TRIGGER framework to adjust negative sampling in the training process such that the scope of false negatives dynamically updates according to the model capacity. Extensive experiments on two benchmarks verify the effectiveness of our approach.

## 1 Introduction

Recently, retrieval-based dialogue draws rising interest from the NLP community (Liu et al., 2022; Lee et al., 2022; Tao et al., 2023; Feng et al., 2023), since it is a promising way towards intelligent human-machine dialogue. Moreover, the technologies in building it show great potential in various applications such as tasked-oriented dialogue assistant (Shu et al., 2022), conversational recommendation (Li et al., 2018) or the recently released interactive large language model (OpenAI, 2022). In this study, we focus on the core of a retrieval dialogue system, i.e., the multi-turn response selection task, which aims to retrieve the best response (golden response) from a pre-defined candidate pool given a dialogue context.

For improving the discriminating power of a retrieval dialogue system, the key is the construction of the candidate pool, or the selection of the negative examples. Since the trivial solution, i.e., randomly sampling utterances from the entire training set, results in too simple and informative negatives (Li et al., 2019), a large body of previous works (Lin et al., 2020; Su et al., 2020; Penha and Hauff, 2020) discuss how to excavate hard negatives that are lexically or semantically similar to the golden response and therefore hard to differentiate. However, due to the inherent one-to-many property of open-domain dialogue, sometimes hard negatives are in fact positive in disguise, or false negatives (Gupta et al., 2021; Lee et al., 2022) and are detrimental to the convergence of retrieval model (Xiong et al., 2020; Zhou et al., 2022).

To verify this point, we perform a pilot study (Section 2) to find that previous negative sampling methods do yield a portion of false negatives and the ratio is higher than random sampling. It is easy to understand since previous methods tend to overlook and thus have little control over the false negative issue, except that Gupta et al. (2021) try to filter out the false negatives in a heuristic manner. Nevertheless, heuristically filtering out negatives that are similar to the golden response is far from enough, and mitigating the false negative issue is a non-trivial problem:

A major challenge is that a randomly sampled utterance could also be more or less appropriate, although it may differ from the golden response in many aspects. Owing to the one-to-many nature (Zhao and Kawahara, 2021; Towle and Zhou, 2022), there usually exists more than one possible dialogue flow, with each flow reflecting different transitions in dialogue topic, user emotion and many other characteristics. To recognize and mitigate false negatives, it is required that we capture the diverse potential transition logic of multiple characteristics in open-domain dialogue, which is

---

[†]Tingchen Fu and Xueliang Zhao contribute equally to this work. This work was done during their internship at Tencent AI Lab.
[*]Corresponding author: Rui Yan (ruiyan@ruc.edu.cn).

difficult to acquire (Xu et al., 2021).

Another challenge lies in the balance between excavating hard negatives and removing false negatives (Cai et al., 2022; Yang et al., 2022). Specifically, we may largely avoid false negatives by always selecting naive negative examples that are obviously inappropriate, which is however uninformative and useless. In other words, the dividing line between the false negatives and the hard negatives is bounded to change dynamically in the training process according to the retrieval model capacity. Though there exist some works employing curriculum learning (Su et al., 2020; Penha and Hauff, 2020), their adjustment of negative sampling is performed in an empirical way independent of the model capacity.

In this research, to cope with the first challenge, we propose to decompose the characteristics in multi-turn conversation into multiple dimensions and represent each with a latent label respectively. To achieve this, we design a sequential variational ladder auto-encoder (SVLAE) to model the transition logic of multiple characteristics and disentangle each other. For the second challenge, we update the negative sampling dynamically in the training process in pace with the optimization of the retrieval model. Specifically, we propose a TRIGGER (TRansItion loGic auGmentEd Retrieval) framework, which consists of T-step and R-step and optimizes the negative sampling process and retrieval model in two steps iteratively.

To summarize, our contributions are three-fold:

(1) We devise a sequential variational ladder auto-encoder to model the multiple orthogonal characteristics in a compositional and disentangled way.

(2) We propose a TRIGGER framework that combines the updating of negative sampling together with the optimization of a retrieval model such that the criterion for negative sampling dynamically changes to pace with the capacity of the retrieval model.

(3) Extensive experiments on two benchmarks verify that when combined, our method significantly improves the existing retrieval models by a large margin and achieves a new state-of-the-art.

## 2 Pilot Study On False Negatives

In this pilot study, we conduct a human evaluation to investigate false negatives hidden in the candidate pool, where the negative samples in the candidate pool are constructed by random sampling or

| Model | False Negative Ratio |
|---|---|
| Random | 3.29 |
| Semi (Li et al., 2019) | 7.16 |
| CIR (Penha and Hauff, 2020) | 7.03 |
| Grey (Lin et al., 2020) | 7.42 |
| HCL (Su et al., 2020) | 6.95 |
| Mask-and-fill (Gupta et al., 2021) | 5.63 |

Table 1: The false negative ratio (%) of selected negative candidates on Douban dataset.

| Model | False Negative Preventing | Negative Updating |
|---|---|---|
| Semi (Li et al., 2019) | ✗ | ✗ |
| CIR(Penha and Hauff, 2020) | ✗ | ✔ |
| HCL(Su et al., 2020) | ✗ | ✔ |
| Grey(Lin et al., 2020) | ✗ | ✗ |
| Mask-and-fill(Gupta et al., 2021) | ✔ | ✗ |
| Ours | ✔ | ✔ |

Table 2: Comparison with other negative sampling methods

four previous methods. Specifically, we ask five in-house annotators to independently scan through negative responses excavated from previous negative sampling methods and mark out the appropriate and acceptable ones. Negatives marked by more than half of the annotators are identified to be false negatives. The experiment is performed on 200 dialogue context sampled from Douban (Wu et al., 2017) and for every context we sample 5 negatives from each sampling method.

The experiment results are shown in Table 1. We can see that compared with random sampling, HCL (Su et al., 2020), CIR (Penha and Hauff, 2020), Grey (Lin et al., 2020) and Semi (Li et al., 2019) have substantially more false negatives. Although mask-and-fill (Gupta et al., 2021) partially solves the problem thanks to its semantic limitation mechanism, it still has a higher mislabel ratio than random sampling.

## 3 Related Work

The task of **multi-turn response selection** aims at selecting a response to match the human input from a large candidate pool (Lowe et al., 2015; Yan et al., 2016; Zhou et al., 2016; Wu et al., 2017; Zhou et al., 2018; Tao et al., 2019a; Jia et al., 2020). Close related to conversational recommendation (Li et al., 2018) and interactive large language model (OpenAI, 2022), it has extensive application in the commercial area. With the recent huge success of

PLMs (Devlin et al., 2018; Liu et al., 2019), post-training PLMs with diverse self-supervised tasks become a popular trend and achieve impressive performance (Xu et al., 2020; Gu et al., 2020; Whang et al., 2021; Han et al., 2021; Fu et al., 2023).

Apart from designing new architecture or new self-supervision task, another branch of work put emphasis on **negative sampling**. Namely, since the quality of negative candidates has a great influence on the retrieval model, a large body of work curate hard negative candidate by searching within the corpus (Su et al., 2020), synthesizing from language model (Gupta et al., 2021) or a combination of both (Lin et al., 2020). However, most previous methods pay little attention to preventing false negatives or updating the negative sampling to adapt with the optimization of the retrieval model, albeit the semantic limitation in Gupta et al. (2021) and the curriculum learning in Penha and Hauff (2020) and Su et al. (2020). The comparison of our methods against previous ones is shown in Table 2.

# 4 Methodology

**Problem Formulation.** Given a dialogue context $c = (u_1, u_2, \cdots, u_N)$ with $u_i$ denoting the $i$-th utterance and $N$ is the number of turns, the objective of a retrieval model $D(\cdot|c, \mathcal{R})$ is to find the golden response $r^+$ from a candidate pool $\mathcal{R} = \{r^+, r_1^-, \ldots, r_n^-\}$ where $\{r_1^-, \ldots, r_n^-\}$ are $n$ negative samples. To enable a retrieval model to differentiable multiple candidates and pick out the $r^+$, the core is the meticulous construction of the candidate pool $\mathcal{R}$.

**Overview.** The proposed approach has two stages: (1) transition logic estimation and (2) dynamic negative updating. In the first stage, we train a transition model to capture the transition logic of multi-level characteristics in a conversation. The transition model is used for detecting characteristics in multiple facets to decide whether a candidate utterance is a potential false negative. However, with the growing model capacity, the criteria for potential false negatives should change accordingly. So in the second stage, we introduce a policy network to determine the negative sampling criteria regarding multi-facet characteristics according to the feedback from the retrieval model. In this way, the negative sampling process paces with the evolution of the retrieval model. At the test stage, the transition model and the policy network are discarded so it causes no extra latency.

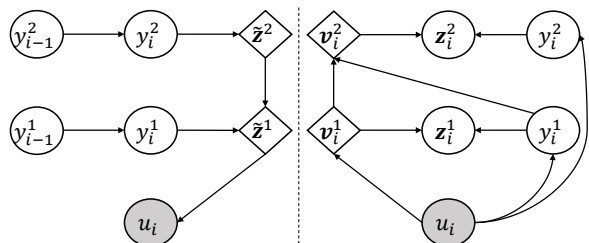

Figure 1: The generation (left) and inference (right) of SVLAE architecture with $L = 2$.

## 4.1 Transition Logic Estimation

To represent and learn the transition logic, we describe each utterance in a dialogue as generated by both a discrete latent label $y$ and a continuous latent feature $\mathbf{z}$ following Kingma and Welling (2013). Because the transition logic is influenced by multiple (orthogonal) factors including but not limited to the dialogue topic, dialogue acts, or speaker's emotion, it would be unwieldy to enumerate each combination of these factors with a single $y$, which would lead to the hypothesis space of $y$ scaling exponentially with the number of factors considered.[1] Instead, inspired from Zhao et al. (2022) and Falck et al. (2021), we propose a Sequential Variational Ladder Auto-Encoder (SVLAE) to model the multiple characteristics in a disentangled and compositional way. The generation and inference in our SVLAE architecture are shown in Figure 1 and elaborated as below.

**Generation** Specifically, we use $L$ latent labels $y^{1:L}$ and latent features $\mathbf{z}^{1:L}$ to describe the characteristics of an utterance in $L$ different facets. In our probabilistic framework, to generate a multi-turn dialogue with $N$ utterances, we first sample $N$ latent labels for every facet $l \in \{1, 2, \ldots L\}$, and then sample corresponding $z^l$ from a mixture of

---

[1] For example, if we consider $L$ facets with $K^l$ the number of possible values for the $l$-th facet, a "single partition" model needs to encode $K^1 \times K^2 \times \ldots \times K^l$ possibilities with a single $y$.

Gaussian[2]:

$$p(y_{1:N}^l) = p(y_1^l) \prod_{i=2}^{N} p_\theta(y_i^l \mid y_{<i}^l)$$

$$p(\mathbf{z}_{1:N}^l) = \prod_{i=1}^{N} p(\mathbf{z}_i^l \mid y_i^l) \qquad (1)$$

$$= \prod_{i=1}^{N} \mathcal{N}(\mathbf{z}^l \mid \mu_\theta(y^l), \Sigma_\theta(y^l))$$

With all $L$ latent features $\mathbf{z}^{1:L}$, the corresponding $i$-th utterance $u_i$ is generated by:

$$\tilde{\mathbf{z}}_i^l = f_\theta([\mathbf{z}_i^l; \tilde{\mathbf{z}}_i^{l+1}]), l = 1, 2, \ldots, L-1$$

$$p(u_i \mid \mathbf{z}_i^{1:L}) = p(u_i \mid \tilde{\mathbf{z}}_i^1) = \prod_{t=1}^{|u_i|} d_\theta(w_t \mid w_{<t}, \tilde{\mathbf{z}}^1),$$

$$(2)$$

In implementation, $p(y_1^l)$ is a uniform distribution over $K^l$ possible values. $p_\theta$, $\mu_\theta(y^l)$ and $\Sigma_\theta(y^l)$ are multi-layer perceptrons. The sequence modeling $p_\theta(y_i^l \mid y_{<i}^l)$ and the utterance generation $d_\theta(w_t \mid w_{<t}, \tilde{\mathbf{z}}^1)$ are both implemented with light-weight transformer.

More details about the neural parameterization could be found in appendix **??**.

**Inference** For the recognition of latent labels $y^{1:L}$ and latent continuous variables $\mathbf{z}^{1:L}$ for utterance[3] $u$, we propose to factorize the variational posterior $q(\mathbf{z}^{1:L}, y^{1:L} \mid u)$ as

$$q(\mathbf{z}^{1:L}, y^{1:L} \mid u) = q_\phi(\mathbf{z}^{1:L} \mid y^{1:L}, u) \prod_{l=1}^{L} q_\phi(y^l \mid u)$$

$$(3)$$

In this way, the inference for each facet is conducted independently, encouraging the capturing of multi-facet disentangled characteristics.

In implementation, $q_\phi(y^l \mid u)$ is parameterized by a 1-layer GRU and a multi-layer perception. Regarding the inference of $z^{1:L}$, we draw inspiration from Tenney et al. (2019); Niu et al. (2022). These studies find that the representation after lower BERT layers usually encodes basic syntactic information while higher layers usually encode high-level semantic information. In light of this, we prepend a special [CLS] token to each utterance $u$ and employ the encoding of the [CLS]

---

[2]We use superscript to denote the facet and the subscript to denote the utterance turn in the rest of the paper.

[3]we omit the subscript in this paragraph to avoid the clutter of notation

token after each layers to discover multiple diverse features about $u$:

$$\mathbf{z}^l \sim \mathcal{N}(\mu_\phi(\mathbf{v}^l, \boldsymbol{\pi}^l \mathbf{W}^l), \Sigma_\phi(\mathbf{v}^l, \boldsymbol{\pi}^l \mathbf{W}^l))$$

$$\mathbf{v}^{l+1} = f_\phi([\mathbf{h}_{[\text{CLS}]}^{l+1}; \boldsymbol{\pi}^l \mathbf{W}^l]) \qquad (4)$$

where $\boldsymbol{\pi}^l \in \mathbb{R}^{K^l}$ is the distribution of $y^l$ calculated by $q_\phi(y^l \mid u)$ and $\mathbf{h}_{[\text{CLS}]}^l$ is the representation of [CLS] token after $l$ layers.

**Optimization** For optimizing the SVLAE, we exploit the evidence lower bound objective (ELBO) to jointly optimize the generation parameter $\theta$ and inference parameter $\phi$.

$$\sum_{i=1}^{N} \mathbb{E}_{\mathbf{z}_i^{1:L} \sim q} \log p(u_i \mid \mathbf{z}_i^{1:L})$$

$$- \mathbb{E}_{y_i^{1:L} \sim q} \text{KL}(q(\mathbf{z}_i^{1:L} \mid y_i^{1:L}, u_i) \| \prod_{l=1}^{L} p(z_i^l \mid y_i^l))$$

$$- \sum_{l=1}^{L} \text{KL}(q(y_i^l \mid u_i) \| p(y_i^l)),$$

$$(5)$$

where KL denotes the Kullback-Leibler divergence. After the training process, the transition model is fixed and used for inferring the latent label of all utterances in the training corpus.

### 4.2 Dynamical Negative Updating

We assume that utterances sharing more latent labels with $r^+$ are more likely to be false negatives and thus should be excluded from $\mathcal{R}$ at the beginning. But with the training process proceeding, the retrieval model gradually acquires the ability to discern the subtle differences in multiple facets and the exclusion criterion should also update. To achieve this, we develop a TRIGGER framework that updates the negative sampling in pace with the retrieval model in two iterative steps:

**T-step** At T-step, we introduce a policy network for predicting the characteristics of the utterance that is most suitable to be negative samples given the current model capacity. Then the policy network receives a reward from the retrieval model to update its parameter.

Specifically, the policy network $\mathcal{P}$ takes the predicted latent label of the golden response $p_\theta(y_{N+1}^{1:L})$ as input and predicts the latent label distribution of the suitably difficult negatives, denoted as $\boldsymbol{\pi}(y^{1:L})$. Then we sample the latent labels $\tilde{y}_1 \sim \boldsymbol{\pi}(y^1), \tilde{y}_2 \sim \boldsymbol{\pi}(y^2), \ldots, \tilde{y}_n \sim \boldsymbol{\pi}(y^n)$ and

| Dataset | Ubuntu (Lowe et al., 2015) | | | Douban (Wu et al., 2016) | | |
|---|---|---|---|---|---|---|
| | Train | Validation | Test | Train | Validation | Test |
| # context-response pairs | 1M | 500K | 500K | 1M | 50K | 6670 |
| # candidates per context | 2 | 10 | 10 | 2 | 10 | 10 |
| Avg #turns per context | 10.13 | 10.11 | 9.80 | 6.69 | 6.75 | 6.45 |
| Avg #words per turn | 13.86 | 17.14 | 17.18 | 15.58 | 14.71 | 16.56 |

Table 3: Statistics of two datasets used in our experiments.

construct a set $S(\tilde{y}_1, \tilde{y}_2, \ldots, \tilde{y}_L)$ consisting of all utterances that are predicted to have exactly the same latent label $\tilde{y}^1, \tilde{y}^2, \ldots, \tilde{y}^L$. We sample negative $r^-$ from the set $S(\tilde{y}_1, \tilde{y}_2, \ldots, \tilde{y}_L)$, and optimize the policy network $\mathcal{P}$ with policy gradient (Sutton et al., 2000):

$$\mathbb{E}_{r^- \sim S(\tilde{y}_1, \tilde{y}_2, \ldots, \tilde{y}_n), \tilde{y}_i \sim \pi(y_i)} D(\tilde{r}^- \mid c, r^+) \quad (6)$$

Intuitively, the reward for the policy network $D(\tilde{r}^- \mid c, r^+)$ is the probability that the retrieval model regards the sampled $r^-$ as a better candidate than $r^+$. In implementation, the policy network is a lightweight bi-directional transformer. More parameterization details could be found in Appendix **??**.

**R-step** At R-step, the retrieval model $D$ is trained to discriminate the golden response $r^+$ apart from the negatives excavated by the policy network. In detail, after the T-step is finished, the parameter of the policy network is fixed and we reconstruct the set $S(\tilde{y}_1, \tilde{y}_2, \ldots, \tilde{y}_L)$ with the updated policy network. Note that it is possible that the safe set is an empty set. If this is the case, we re-sample $y^{1:L}$ from the sampling policy $\pi(y^{1:L})$. The objective of the retrieval model is thus:

$$\max_D \mathbb{E}_{\tilde{r}^- \in C^-} D(r^+ \mid c, \{r^-\} \cup r^+). \quad (7)$$

Compared with the original training objective, the new one in Eq. 7 restricts the scope of negatives to the set $S(\tilde{y}_1, \tilde{y}_2, \ldots, \tilde{y}_L)$, thus mitigating the false negatives and rendering the selection of negatives pace with the optimization of the retrieval model.

A high-level algorithm for our proposed framework is shown in Algorithm 1.

## 5 Experiment

### 5.1 Datasets

We conduct experiments on two benchmarks: Ubuntu Corpus V1, and Douban Corpus. The statistics of these three datasets are shown in Table 3.

---

**Algorithm 1** The proposed TRIGGER framework.

1: **Input:** A retrieval model $D$, training corpus, maximum training step for the transition model and retrieval model $M_1$ and $M_2$.
2: **for** $m \leftarrow 1$ to $M_1$ **do**
3:     Sample a mini-batch $(c, r)$ from the training corpus.
4:     Recognize the latent labels and latent features with the inference model $\phi$.
5:     Generate the conversation with generation model $\theta$.
6:     Optimize the SVLAE with the objective in Eq. 5.
7: **end for**
8: **for** $m \leftarrow 1$ to $M_2$ **do**
9:     {An new episode begins.}
10:     Sample a mini-batch of $(c, r)$ training corpus.
11:     Compute the latent label distribution of the suitably difficult negatives $\pi(y^{1:L})$ and construct the set $S(\tilde{y}_1, \tilde{y}_2, \ldots, \tilde{y}_L)$
12:     Sample $\{r^-\}$ from the set to compose $\mathcal{R}$.
13:     **if** m is odd **then**
14:         {T-step}
15:         Optimize the policy network $\mathcal{P}$ with Eq. 6
16:     **else**
17:         {R-step}
18:         Optimize the retrieval model $\mathcal{D}$ with Eq. 7
19:     **end if**
20:     {An episode ends.}
21: **end for**
22: **Return:** the retrieval model $D$.

---

**Ubuntu Corpus V1** (Lowe et al., 2015) is a multi-turn response selection dataset in English collected from chatting logs, mainly about seeking technical support for problems in using the Ubuntu system. We use the copy shared by (Xu et al., 2016), which replaces all the numbers, URLs and paths with special placeholders.

**Douban Corpus** (Wu et al., 2016) is an open-domain Chinese dialogue dataset from the Douban website, which is a popular social networking service. Note that for the test set of Douban corpus, one context could have more than one correct response as the golden response is manually labeled.

### 5.2 Evaluation Metrics

Following previous works (Tao et al., 2019b; Xu et al., 2020), we use *recall* as our evaluation metrics. The recall metric $R_{10}@k$ means the correct response is within the top-$k$ candidates scored by the retrieval model out of 10 candidates in total.

| Model | Ubuntu | | | Douban | | | | |
|---|---|---|---|---|---|---|---|---|
| | $R_{10}@1$ | $R_{10}@2$ | $R_{10}@5$ | MAP | MRR | P@1 | $R_{10}@1$ | $R_{10}@2$ |
| BERT (Devlin et al., 2018) | 80.8 | 89.7 | 97.5 | 59.1 | 63.3 | 45.4 | 28.0 | 47.0 |
| +Semi (Li et al., 2019) | 81.0 | 89.8 | 97.5 | 60.3 | 63.8 | 46.0 | 28.2 | 47.9 |
| +CIR (Penha and Hauff, 2020) | 81.2 | 89.9 | 97.6 | 60.7 | 64.2 | 46.4 | 28.5 | 49.1 |
| +Gray (Lin et al., 2020) | 81.5 | 90.1 | 97.5 | 61.5 | 64.8 | 47.1 | 29.1 | 50.7 |
| +HCL (Su et al., 2020) | 81.5 | 90.2 | 97.6 | 61.7 | 65.9 | 48.0 | 30.4 | 51.0 |
| +MF (Gupta et al., 2021) | 81.6 | 90.0 | 97.5 | 61.9 | 66.0 | 48.1 | 30.5 | 51.3 |
| +TRIGGER (Ours) | $81.9^{\dagger}$ | 90.4 | 97.5 | $62.9^{\dagger}$ | $67.8^{\dagger}$ | $50.2^{\dagger}$ | $32.5^{\dagger}$ | $52.1^{\dagger}$ |
| SA-BERT (Gu et al., 2020) | 85.5 | 92.8 | 98.3 | 61.9 | 65.9 | 49.6 | 31.3 | 48.1 |
| +Semi (Li et al., 2019) | 85.8 | 93.1 | 98.9 | 62.3 | 66.4 | 50.0 | 31.7 | 49.0 |
| +CIR (Penha and Hauff, 2020) | 86.0 | 93.5 | 99.0 | 62.4 | 66.6 | 50.3 | 31.8 | 49.7 |
| +Gray (Lin et al., 2020) | 86.1 | 93.4 | 99.1 | 62.8 | 67.0 | 50.3 | 32.0 | 50.3 |
| +HCL (Su et al., 2020) | 86.7 | 94.0 | 99.2 | 63.9 | 68.1 | 51.4 | 33.0 | 53.1 |
| +MF (Gupta et al., 2021) | 86.9 | 93.7 | 98.9 | 63.6 | 68.4 | 52.0 | 33.5 | 52.9 |
| +TRIGGER (Ours) | $87.1^{\dagger}$ | 93.8 | 98.7 | 64.1 | $69.5^{\dagger}$ | $52.9^{\dagger}$ | $34.3^{\dagger}$ | $53.7^{\dagger}$ |
| BERT-FP (Han et al., 2021) | 91.1 | 96.2 | 99.4 | 64.4 | 68.0 | 51.2 | 32.4 | 54.2 |
| +Semi (Li et al., 2019) | 91.2 | 96.2 | 99.3 | 65.8 | 68.7 | 53.6 | 34.3 | 54.7 |
| +CIR (Penha and Hauff, 2020) | 91.2 | 96.4 | 99.3 | 66.5 | 70.6 | 54.8 | 35.5 | 55.0 |
| +Gray (Lin et al., 2020) | 91.3 | 96.3 | 99.4 | 66.7 | 70.5 | 55.0 | 36.3 | 54.8 |
| +HCL (Su et al., 2020) | 91.2 | 96.3 | 99.4 | 67.1 | 70.9 | 55.3 | 36.4 | 55.7 |
| +MF (Gupta et al., 2021) | 91.3 | 96.4 | 99.3 | 67.2 | 71.4 | 55.4 | 36.2 | 56.7 |
| +TRIGGER (Ours) | $\mathbf{91.7}^{\dagger}$ | **96.6** | 99.4 | $\mathbf{67.9}^{\dagger}$ | $\mathbf{72.4}^{\dagger}$ | $\mathbf{56.5}^{\dagger}$ | 36.7 | $\mathbf{59.5}^{\dagger}$ |

Table 4: Evaluation results on the test sets of the Ubuntu and Douban. Numbers in bold are best results. $^{\dagger}$ denotes that the improvement over the most competitive baseline is statistically significant (t-test, p-value <0.05)

We use $R_{10}@1$, $R_{10}@2$ and $R_{10}@5$ in our experiment. As mentioned above, Douban Corpus contains more than one positive response from the candidates: we also measure *MAP* (mean average precision), *MRR* (mean reciprocal rank) and *P@1* precision at one.

## 5.3 Implementation Details

Our method is implemented by Pytorch and performed on 2×24 GiB GeForce RTX 3090. The code is implemented with Hugging Face[4] and the code is available at https://github.com/TingchenFu/EMNLP23-LogicRetrieval.

For hyper-parameter selection in the transition model, retrieval model, and policy network, we sweep the learning rate among $[5e-6, 1e-5, 2e-5, 4e-5, 5e-5]$ and sweep the batch size among $[4, 8, 16, 32]$ for each dataset. We only keep the last 15 turns of a dialogue context and the maximum length of a context-candidate pair is 256. The gradient is clipped to 2.0 to avoid the gradient explosion. All models are learned with Adam (Kingma and Ba, 2015) optimizer with $\beta_1 = 0.9$ and $\beta_2 = 0.999$. An early stop on the validation set is adopted as a regularization strategy. We report the averaged performance over three repetitive experiments for our method.

For transition model, $p_\theta(y_i^l \mid y_{<i}^l)$ is implemented as a transformer. Similar to word embedding, we obtain the embedding of $y_1^l, y_2^l, \ldots, y_N^l$

[4]https://www.huggingface.co

by looking up a randomly initialized and learnable embedding matrix, and then the embeddings are modeled by a uni-directional 3-layer transformer.

Similarly, $d_\theta(w_t \mid w_{<t}, \tilde{\mathbf{z}}^1)$ is implemented as a 6-layer uni-directional transformer. The $\tilde{\mathbf{z}}^1$ is mapped to the same dimension as the hidden representation of the transformer with a learnable matrix before prepending as a special token to the word embedding of $u$.

For inference the latent labels given an utterance, $q_\phi(y^l \mid u)$ is composed of a 1-layer GRU and a multi-layer perceptron. The former encodes an utterance $u$ into a dense vector $\mathbf{u}$ while the latter maps the dense vector to a $K^l$-way categorical distribution.

The policy network is a bi-directional transformer together with $L$ dense matrices $\mathbf{W}_{1:L}^p$. The matrices map $p_\theta(y_{N+1}^{1:L})$ into $L$ vectors in the same dimension before concatenating them together as an embedding sequence and input into the transformer. The hidden representation after the last layer of transformer is then mapped back to the dimension $K^1, K^2, \ldots, K^L$,

## 5.4 Retrieval Models

As a negative sampling approach, our proposed TRIGGER framework is orthogonal to the retrieval models and theoretically our approach can be combined with any dialogue retrieval model seamlessly. To validate the universality of our approach, we perform experiments on the following base retrieval

model:

**BERT** (Devlin et al., 2018) is the vanilla BERT model fine-tuned on the response selection task with no post-training. **SA-BERT** (Gu et al., 2020) enables the BERT model to be aware of different speakers in the concatenated context sequence by applying speaker position embedding. **BERT-FP** (Han et al., 2021) uses fine-grained post-training to help the Bert model distinguish the golden response and negatives from the same dialogue session.

### 5.5 Baselines

To verify the effectiveness of our approach, we draw a comparison with the following baseline method on negative sampling methods: **Semi** (Li et al., 2019) re-scores negative samples at different epochs to construct new negatives for training. **CIR** (Penha and Hauff, 2020) exploits curriculum learning and transits from easy instances to difficult ones gradually. **Gray** (Lin et al., 2020) trains another generation model to synthesize new negatives. **HCL** (Su et al., 2020) proposes an instance-level curriculum and a corpus-level curriculum with a pacing function to progressively strengthen the capacity of a model. **Mask-and-fill** (Gupta et al., 2021) propose a mask-and-fill approach that simultaneously considers the original dialogue context as well as a randomly picked one to synthesize negative examples.

### 5.6 Experiment Results

The quantitative results are shown in Table 4, We can observe that our method significantly improves the original base retrieval models on most metrics, showing the universality and robustness of our method. When combined with BERT-FP, our method achieves the new state-of-the-art for most metrics, with an absolute improvement of $0.6\%$ and $4.3\%$ on $R_{10}@1$ for Ubuntu benchmark and Douban benchmark respectively.

### 6 Analysis

Apart from the overall performance, we are particularly curious about and make further analysis to understand the following questions: **Q1**: How does each component and mechanism contribute to the overall performance? **Q2**: How does the lexical similarity between the context and the response influence the performance of our method? **Q3**: How is the robustness of our method under adversarial

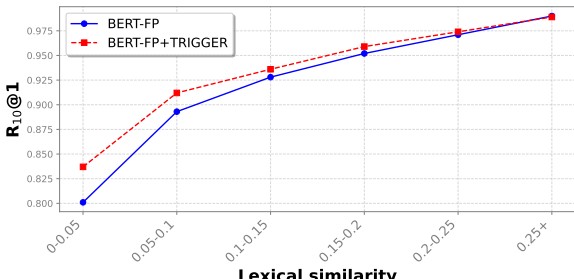

Figure 2: Retrieval Performance vs. lexical similarity on Ubuntu.

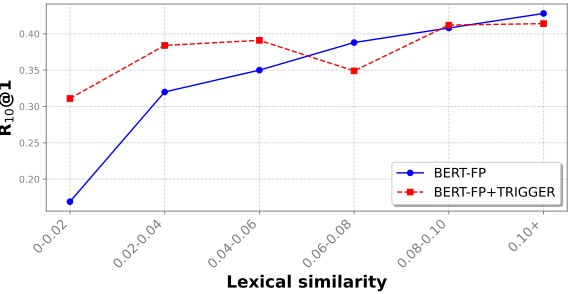

Figure 3: Retrieval Performance vs. lexical similarity on Douban.

or noisy input? **Q4**: How can we understand and interpret the learned transition logic?

### 6.1 The Impact of Each Component

**Answer to Q1** To study how each component works, we conduct an ablation study mainly considering the following variants: *-transition*: The transition-aware hard negatives are replaced with randomly sampled negatives; *-update*: The T-step is removed the transition-aware hard negatives are fixed during the training process; *-feature*: The latent features $\mathbf{z}^{1:L}$ are removed from the SVLAE architecture. *-label*: The latent labels $y^{1:L}$ are removed from the SVLAE architecture.

The experiment results are shown in Table 5. From the table, we could observe that: (1) There is an evident degradation when the meticulously excavated negatives are replaced with randomly sampled ones This suggests that our negative sampling methods are crucial to the retrieval performance, and that simply increasing the number of negative examples may not be sufficient. (2) The updating of the sampling policy in T-step plays an important role as its removal of it causes an obvious drop. This result justifies the necessity of updating hard negatives dynamically. (3) Both the latent labels $y^{1:L}$ and latent features $\mathbf{z}^{1:L}$ contribute to the model performance. This is likely because

| Models | Ubuntu | | | Douban | | | | |
|---|---|---|---|---|---|---|---|---|
| | $R_{10}@1$ | $R_{10}@2$ | $R_{10}@5$ | MAP | MRR | P@1 | $R_{10}@1$ | $R_{10}@2$ |
| BERT-FP | 91.1 | 96.2 | 99.4 | 64.4 | 68.0 | 51.2 | 32.4 | 54.2 |
| +TRIGGER (Ours) | 91.7 | 96.6 | 99.4 | 67.9 | 72.4 | 56.5 | 36.7 | 59.5 |
| *-transition* | 90.9 | 95.9 | 99.2 | 64.5 | 68.2 | 51.5 | 32.6 | 54.8 |
| *-update* | 91.2 | 96.3 | 99.3 | 65.9 | 70.4 | 53.8 | 34.9 | 56.3 |
| *-label* | 91.5 | 96.4 | 99.4 | 67.3 | 72.0 | 55.9 | 36.1 | 58.3 |
| *-feature* | 91.4 | 96.3 | 99.4 | 67.0 | 71.6 | 54.2 | 35.5 | 57.4 |

Table 5: Results of ablation study on two benchmarks.

| Rank | Transition | Probability |
|---|---|---|
| 1 | 233→115 | 14.25% |
| 2 | 265→211 | 12.90% |
| 3 | 211→265 | 10.45% |
| 4 | 274→210 | 2.20% |
| 5 | 110→274 | 2.22% |

Table 6: The top-5 2-gram transition pairs in Ubuntu and their transition probability.

| Rank | Latent Label | Implication |
|---|---|---|
| 1 | 233 | Short instructions about paste operation. |
| 2 | 115 | Reference to a URL. |
| 3 | 211 | Audio and sound configuration in Ubuntu. |
| 4 | 265 | Sound system breakdown. |
| 5 | 110 | Brief yes/no answer. |

Table 7: The implications of top-5 latent label at facet $L$ in Ubuntu.

they provide complementary information that can be used to learn the transition logic.

## 6.2 The Impact of Lexical Similarity

**Answer to Q2**. To have a better understanding of the impact of lexical similarity, we bin the test set of Ubuntu and Douban into different bins according to the similarity between the context and the golden response[5]. The improvement on BERT-FP (Han et al., 2021) is shown in Figure 2 and Figure 3. We could see that our method is helpful and substantially improve the retrieval performance on most bins. The improvement is obvious, especially in the harder scenario where the context and the golden response are less similar. We attribute the gains to the transition logic, which exempts the retrieval model from the interference of false negatives.

## 6.3 Performance under Adversarial Attack

**Answer to Q3:** Reducing false negatives can effectively mitigate the noise in supervision signal and therefore stabilize the training process (Zhou

---

[5]measured in uni-gram F1

et al., 2022). In addition, it might render the model more robust to adversarial input or noisy input in real-world applications. To verify this point, inspired by Jia and Liang (2017); Yuan et al. (2019); Whang et al. (2021), we alter the candidate pool of each case by substituting all the negative candidates with the utterances in the dialogue context, which is a more challenging setting than previous scenarios (Whang et al., 2021). The experiment results on Ubuntu and Douban are shown in Figure 2 and Figure 3 respectively.

According to the experiment results, all three base models deteriorate severely under our attack. It also reveals that relying on superficial cues is in fact a common phenomenon in PLM-based retrieval models, not limited to BERT-FP (Han et al., 2021). In comparison, we could see that the three models are much more robust when combined with our proposed TRIGGER strategy.

## 6.4 Case Study

**Answer to Q4**: In this section, we aim at interpreting what is learned by the transition model in an intuitive way. As a simplification, we only consider the "2-gram" transition of latent labels at the $L$-th facet (The most abstract one). Specifically, we first recognize the latent label at facet $L$ for all utterances in the corpus with our transition model. Next, we investigate the statistics of latent labels pairs $(y_i^L, y_{i+1}^L)$ appeared in the transition sequence $[y_1^L, y_2^L, \ldots, y_N^L]$ across the entire corpus.

The most frequent "2-gram" transitions in Ubuntu, as well as their transition probability (The percentage that the first state is succeeded by the second one, other than the frequency of the "2-gram"), is shown in Table 6. Besides, we review the 5 most frequent latent labels in facet $L$. By observing randomly sampled 100 utterances with the corresponding latent label from the Ubuntu corpus, we manually induce their implications as shown in Table 7.

| Models | Ubuntu | | | Douban | | | | |
|---|---|---|---|---|---|---|---|---|
| | $R_{10}@1$ | $R_{10}@2$ | $R_{10}@5$ | MAP | MRR | P@1 | $R_{10}@1$ | $R_{10}@2$ |
| BERT(Devlin et al., 2018) | 2.0 | 4.0 | 17.0 | 22.3 | 25.0 | 8.7 | 3.2 | 6.5 |
| BERT+TRIGGER | $13.4^\dagger$ | $20.2^\dagger$ | $48.1^\dagger$ | $37.6^\dagger$ | $42.0^\dagger$ | $23.5^\dagger$ | $13.7^\dagger$ | $21.1^\dagger$ |
| BERT-FP (Han et al., 2021) | 52.5 | 57.4 | 69.2 | 32.5 | 36.8 | 21.9 | 12.9 | 17.3 |
| BERT-FP+TRIGGER | $76.2^\dagger$ | $81.9^\dagger$ | $90.1^\dagger$ | $38.8^\dagger$ | $43.0^\dagger$ | $25.4^\dagger$ | $16.2^\dagger$ | $23.5^\dagger$ |

Table 8: Evaluation results on Ubuntu and Douban under attack. $^\dagger$ denotes that the improvement over the original model is statistically significant (t-test, p-value <0.05).

## 7   Conclusion

In this study, we target at the false negative issue in dialogue retrieval. We recognize that previous negative sampling methods lead to more false negatives than random sampling, which is detrimental to model optimization. So we propose a TRIGGER framework in which we model the inherent transition logic in open-domain dialogue in multiple characteristics with our SVLAE architecture and combine the negative sampling process with the optimization of the retrieval model. In this way, the dividing line between the hard negatives and false negatives is updated dynamically. Extensive experiments verify the efficacy of our approach.

## Limitations

All technologies built upon the large-scale PLM more or less inherit their potential harms (Bender et al., 2021). Besides, we acknowledge some specific limitations within our methods: We only verify the effectiveness of our method on several recent PLM-based methods, but not on early methods without PLM, like SMN (Wu et al., 2017) or ESIM (Chen and Wang, 2019). But since our approach is orthogonal to the base retrieval model, we are promising that our proposal could be easily adapted to these methods.

## Ethical Considerations

This paper will not pose any ethical problems. First, multi-turn response selection is an old task in natural language processing, and several papers about this task are published at EMNLP conferences. Second, all the datasets used in this paper have been used in previous papers. Our method should only be used to boost the performance of the retrieval dialogue system or other research use but not for any malicious purpose.

## Acknowledgement

This work was supported by the National Key Research and Development Program of China (No. 2020YFB1406702), National Natural Science Foundation of China (NSFC Grant No. 62122089), Beijing Outstanding Young Scientist Program NO. BJJWZYJH012019100020098, and Intelligent Social Governance Platform, Major Innovation & Planning Interdisciplinary Platform for the "Double-First Class" Initiative, Renmin University of China, the Fundamental Research Funds for the Central Universities, and the Research Funds of Renmin University of China.

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
