# OpenReview forum: "Logic Unveils Truth, While Disguise Obscures It: Transition Logic Augmented Response Selection for Multi-Turn Dialogue"
_EMNLP/2023/Conference — EMNLP 2023 Findings_

### Official Review · Reviewer_kX1o · 2023-07-28

**Typos Grammar Style And Presentation Improvements:** Formula 2, what is f(theta)? Formula …
**Soundness:** 4

**Excitement:**

3: Ambivalent: It has merits (e.g., it reports state-of-the-art results, the idea is nice), but there are key weaknesses (e.g., it describes incremental work), and it can significantly benefit from another round of revision. However, I won't object to accepting it if my co-reviewers champion it.

**Paper Topic And Main Contributions:**

This paper is addressing the issue of dialogue response retrieval and negative sampling, targeting to filter false negative samples from negative samples.
The author identifies the false negative issue by conducting human annotation and analysis.
The author proposes a sequential variational ladder auto-encoder to capture the transition in dialogue, and further facilitate the identification of false negative responses, which helps the PLM better use negative sampling for training.
Experiment results prove the efficiency of the proposed methods.

**Questions For The Authors:**

Is current transition modelling using a transformer structure with no pre-trained weights or load from any related pre-training models? Why is that?

During the session, is the creation of different difficulty negatives or sets related to different facets? Can you elaborate on the process of sampling negatives from the set?

Negative sampling is a very general technique in information retrieval, is there any discussion for the future work of this proposed method in other areas or domains?

**Reasons To Accept:**

The problem of a false negative in dialogue retrieval is well discussed and the methods are proved through well-designed experiments in a very scientific manner.

Overall, the paper is of high quality with good English, mathematical expression, tables and figures.

Section 2 is really interesting which use human evaluation to identify the situation of false negative, it would be better if we slightly increase the sampling size.

The discussion and results are clear and sound.

**Reasons To Reject:**


Compared with more powerful generative PLM-based methods (especially LLMs), the author fails to justify why dialogue retrieval methods which can only select from existing dialogues instead of generating more diverse or new responses are still meaningful and attractive for the research.
All the related works about dialogue retrieval are from before 2020 and there are only limited words on its application, which makes the reviewer doubts the potential applications of dialogue retrieval in future NLP.


The transition between the problem to your prosed methods in the introduction is not smooth, which makes it hard to understand why this specific encoder or framework is proposed.

**Reproducibility:**

4: Could mostly reproduce the results, but there may be some variation because of sample variance or minor variations in their interpretation of the protocol or method.

**Reviewer Confidence:**

3: Pretty sure, but there's a chance I missed something. Although I have a good feel for this area in general, I did not carefully check the paper's details, e.g., the math, experimental design, or novelty.

---

> ### Author Rebuttal · Authors · 2023-08-29
>
> Thanks for your time and expertise invested in the reviewing process. We are grateful for your appraisal that our manuscript is of high quality and interesting. We carefully read through your insightful comments and we would like to address your concerns one by one.
>
>
> **Q1: Why dialogue retrieval methods are still meaningful and attractive for the research?**
>
> **A1**: Good point! The emergence of large language models in 2022 and 2023, alongside the impressive modeling capabilities of generative dialogue systems (e.g., ChatGPT), has sparked hopes for AGI and raised questions regarding the relevance of retrieval dialogue systems. Nevertheless, we maintain that retrieval dialogue systems warrant further study, with their values and advantages primarily residing in:
>
> + Enhanced safety and control. As responses are chosen from a predefined candidate pool, retrieval dialogue systems offer greater content control, thereby mitigating the presence of toxic words or potentially unsafe content.
>
> + Retrieval-augmented LLM. Many researchers contend that retrieval should play a vital role in next-generation language models to enhance their factuality, interpretability, and trustworthiness. For example, LLMs are less likely to hallucinate when provided with retrieved knowledge from a given knowledge base [1][2][3].
>
> + Retrieval-after-LLM. LLM output can be unstable due to sensitivity to prompts and decoding strategies. A potential solution is to first generate multiple candidates using LLM to simulate various personas [4] or engage in multi-agent debates [5]. A retrieval model may then select the most suitable candidate as a response.
>
> + Reduced computational cost. While LLMs generate high-quality output, the substantial computational expense of APIs (e.g., chatGPT and GPT-4) should not be disregarded, and prior research [6] indicates that LLM (GPT-3) may not be the optimal choice when considering both generation quality and API cost.
>
> + Tool retrieval for LLM. External tools or APIs remain necessary for addressing higher-level tasks with LLMs, and a retrieval system is required to select an appropriate tool (API) from thousands of API documents based on specific human instructions [7]. Given the functional similarity of APIs and the existence of multiple solution paths (chains of API calls), randomly sampled negative APIs may suffer from false negative issues, and our approach offers a potential remedy for this problem.
>
> Therefore, there are still discussions on the techniques of dialogue retrieval.[8][9][10]
>
>
> [1]Check Your Facts and Try Again: Improving Large Language Models with External Knowledge and Automated Feedback, Arxiv 2023.
>
> [2]Enabling large language models to generate text with citations, Arxiv 2023
>
> [3]Rarr: Researching and revising what language models say, using language models.
>
> [4]Unleashing cognitive synergy in large language models: A task-solving agent through multi-persona self-collaboration, Arxiv 2023
>
> [5]Improving factuality and reasoning in language models through multiagent debate, Arxiv 2023.
>
> [6]The economic trade-offs of large language models: A case study, ACL 2023.
>
> [7]ToolLLM: Facilitating Large Language Models to Master 16000+ Real-world APIs, Arxiv 2023.
>
> [8]CORE: Cooperative Training of Retriever-Reranker for Effective Dialogue Response Selection, ACL 2023
>
> [9]A Textual Dataset for Situated Proactive Response Selection, ACL 2023
>
> [10]Towards Efficient Coarse-grained Dialogue Response Selection, ACM Transactions on Information Systems 2023.
>
>
>
>
>
>
> **Q2: Hard to understand why this specific encoder or framework is proposed.**
>
> **A2**: The proposed framework aims at identifying and eliminating the potential false negatives in the negative sampling process. However, because of the one-to-many property in open-domain dialogue, the most appropriate response may vary if we consider different (and orthogonal) factors such as dialogue topic, speaker emotion, and so on, posing a great challenge to the identification of false negatives.
>
> To deal with this, we attempt to model the transition logic of these multiple orthogonal characteristics and predict the characteristic distributions of the golden response. Given the predicted characteristic distribution, we dynamically construct and adjust a negative set $S(\tilde{y}_1, \tilde{y}_2, \ldots, \tilde{y}_L)$. Therefore, the false negatives are diminished and the difficulty of the negatives is updated, constituting our TRIGGER framework.
>
>
>
> **Q3: Is current transition modeling using a transformer structure with no pre-trained weights or load from any related pre-training models? Why is that?**
>
> **A3**: Regarding the inference module, we load weights from a (tiny-)BERT model, as different layers of the BERT model often encode varying levels of semantic abstraction [1], which assists in  acquiring the transition patterns of multiple orthogonal characteristics.
>
> For the generation module, we employ no pre-trained weights but instead use a randomly initialized unidirectional transformer, as transition modeling inherently involves sequence modeling of latent labels, and the latent label sequence does not correspond to any existing language.
>
>
> [1]BERT Rediscovers the Classical NLP Pipeline, ACL 2019.
>
>
> **Q4: During the session, is the creation of different difficulty negatives or sets related to different facets? Can you elaborate on the process of sampling negatives from the set?**
>
> **A4**: Good question! The difficulty of a negative sample/set depends on its/their latent labels in $L$ facets, or $y^{1:L}$. In essence, the more a negative sample resembles the golden response in terms of latent labels, the more probable it is to be a false negative or hard negative.
>
> In detail, given the predicted latent label of the golden response $p_\theta(y^{1:L}_{N+1})$, the policy network $\mathcal{P}$ outputs distribution $\pi(y^1), \pi(y^2), \ldots, \pi(y^L)$, from which we sample $\tilde{y}_1 \sim \pi(y^1), \tilde{y}_2 \sim \pi(y^2), \ldots, \pi(y^L) \sim \tilde{y}_L$. The negative set consists of all the utterances whose latent labels in $L$ facets are exactly $\tilde{y}_1, \tilde{y}_2, \ldots, \tilde{y}_L$.
>
> In summary, the policy network establishes the latent label distribution of the negative set $S$ across $L$ facets, with the similarity between this distribution and the latent label distribution of the golden response determining the negative difficulty. We apologize for any ambiguity in lines 299-310 and will ensure that the details are clarified in the final version.
>
>
> **Q5: Is there any discussion for the future work of this proposed method in other areas or domains?**
>
> **A5**: Thanks for your suggestion! The proposed approach can be easily tranferred to and adapted other subfields:
>
> + conversational search. This approach to information retrieval involves users interacting with an AI assistant in a conversation session to iteratively refine their queries and seek additional information. As it shares the intrinsic features of open-domain dialogue and requires negative sampling during training, our method can be integrated with existing conversational retrieval systems to enhance their performance.
> + conversational recommendation. This system aims to provide accurate product recommendations by directly extracting user preferences from dialogue history. However, like traditional recommendation systems, it necessitates negative sampling for optimization. Given that randomly sampled negatives pose risks of uninformative sampling or false negatives, our approach can be employed to strike a balance between unearthing hard negatives and eliminating false negatives.

---

### Official Review · Reviewer_oe9s · 2023-08-04

**Soundness:** 3

**Excitement:**

2: Mediocre: This paper makes marginal contributions (vs non-contemporaneous work), so I would rather not see it in the conference.

**Paper Topic And Main Contributions:**

This paper investigates the false positive problem in multi-turn response selection. The paper proposes a sequential variational auto-encoder to capture the diverse one-to-many transition pattern of multiple characteristics in open-domain dialogue and a TRIGGER framework that combines the updating of negative sampling together with the optimization of a retrieval model.  Experimental results show the effectiveness of the proposed method.

**Reasons To Accept:**

1. The paper is easy to follow.
2. The method is evaluated on multiple datasets and shows better performance than baselines.

**Reasons To Reject:**

1. The false negative problem is already extensively studied, and the idea of modeling the inherent transition logic is not new.
2. The proposed method is complex, and the improvement over baselines is marginal. The ablation study shows little difference between the method and its several variants.
3. Large language models such as Chatgpt should perform this task well. What are their results?

**Reproducibility:**

3: Could reproduce the results with some difficulty. The settings of parameters are underspecified or subjectively determined; the training/evaluation data are not widely available.

**Reviewer Confidence:**

4: Quite sure. I tried to check the important points carefully. It's unlikely, though conceivable, that I missed something that should affect my ratings.

---

> ### Author Rebuttal · Authors · 2023-08-29
>
> Thank you for your constructive comments and suggestions, and they are exceedingly helpful for us to improve our paper. We carefully read through your comments and we would like to address your concerns point-by-point in below:
>
>
> **Q1: The false negative problem is already extensively studied**
>
> **A1**: We appreciate your guidance and recognize the extensive body of literature addressing negative sampling in the field of multi-turn response selection, as discussed in Section 3. Nonetheless, our comparison with prior negative sampling methods reveals that they either inadequately address the false negative issue or fail to implement negative updating (Table 2), thus inspiring our research. Furthermore, the human evaluation presented in Table 1 indicates that this problem remains unresolved.
>
>
> **Q2: The idea of modeling the inherent transition logic is not new.**
>
> **A2**: Thanks for your comment. The novelty of our proposed Sequential Variational Ladder Auto-Encoder (SVLAE) lies in encoding the transition of multiple characteristics in a disentangled way with a ladder-like architecture, as described in Section 4.1 and illustrated in Figure 1. The motivation behind the proposed approach is to model the dynamics of multiple orthogonal characteristics, thereby identifying possible false negatives that conform to the transition logic in one or more facets. To the best of our knowledge, we are the first to devise this variational and ladder-like architecture for negative sampling in dialogue retrieval. We are sorry for the caused ambiguity and we will definitely highlight the novelty more prominently in the final version.
>
>
>
> **Q3: The proposed method is complex, and the improvement over baselines is marginal.**
>
> **A3**: Thank you for your remarks. We recognize that our approach may not yield a sharp enhancement in the retrieval performance of the Douban or Ubuntu benchmark, given their inherent dataset complexity and the limitations of the base retrieval model. However, as demonstrated in Table 4, the significant test (student test) reveals that TRIGGER notably surpasses the most competitive baseline method. Additionally, it is crucial to note that our proposed TRIGGER framework significantly bolsters the robustness of retrieval models, as evidenced by Table 6, which is vital for deployment. Besides, our method is compatible with any base retrieval model and can be integrated with more advanced PLM encoders to further augment performance
>
>
> **Q4: The ablation study shows little difference between the method and its several variants.**
>
> **A4**: Your observation is correct. In the development of our method, we also note that there is little difference between ours and the *-label* variant. In fact, the latent label $y^{1:L}$ is mainly used for categorizing the utterances in the corpus into different groups and constructing the negative set $S(\tilde{y}_1,\tilde{y}_2,\tilde{y}_3,\ldots,\tilde{y}_L)$. Furthermore, the discrete $y^{1:L}$ facilitates the interpretation of the discovered transition logic, making it more explainable.
>
> We perform a significant test on our ablation experiments and the results are shown as below:
> |Variant | R$_{10}$@1 | R$_{10}$@2 | R$_{10}$@5 | MAP | MRR| P@1 | R$_{10}$@1 | R$_{10}$@2 |
> |------|------|------|------|------|------|------|------|------|
> |-transition | 90.9* | 95.9* | 99.2 | 64.5* | 68.2*  | 51.5* |32.6* | 54.8* |
> |-update |91.2* |96.3| 99.3| 65.9*| 70.4* |53.8* |34.9*| 56.3*|
> |-label |91.5| 96.4| 99.4| 67.3| 72.0| 55.9*| 36.1*| 58.3*|
> |-feature| 91.4*| 96.3| 99.4| 67.0*| 71.6*| 54.2*| 35.5*| 57.4*|
>
> The numbers marked with an asterisk are significantly (t-test, $p<0.05$) inferior to the TRIGGER.
>
>
> **Q5: Large language models such as Chatgpt should perform this task well. What are their results?**
>
> **A5**: Thanks for your advice. We perform an experiment with the ChatGPT with the following prompt:
>
> "Given the following conversation between two speakers:
>
> speaker A: {}
>
> speaker B: {}
>
> speaker A: {}
>
> speaker B: {}
>
> ......
>
> and the 10 candidate replies numbered from A to J:
>
> reply A: {}
>
> reply B: {}
>
> reply C: {}
>
> ......
>
> replay J: {}
>
> which candidate reply is a good response to the above dialogue context to form into a fluent, coherent and consistent conversation?
>
> Please output a rank of above 10 candidates to indicate whether the candidate reply is appropriate.
>
> The more highly ranked the response, the more appropriate it is. For example, 'BDCAIJGHFE' means B is the best response, followed by D,C,A ... while E is the worst one.
>
> Only output the rank in a single line. Do not explain. "
>
> We randomly sample 100 cases (1000 context-response pairs) from the Ubuntu benchmark The experiment results are shown below.
> |    |R$_{10}$@1 | R$_{10}$@2 | R$_{10}$@5 |
> |----|----|----|----|
> |GPT-3.5-turbo | 0.31 | 0.46 | 0.82 |
>
> From the table, we can observe that ChatGPT is not the final solution for the multi-turn response selection task.

---

### Official Review · Reviewer_wMUd · 2023-08-05

**Typos Grammar Style And Presentation Improvements:** 1. Further definition of "model capac…
**Soundness:** 4

**Excitement:**

3: Ambivalent: It has merits (e.g., it reports state-of-the-art results, the idea is nice), but there are key weaknesses (e.g., it describes incremental work), and it can significantly benefit from another round of revision. However, I won't object to accepting it if my co-reviewers champion it.

**Missing References:**

[1]: RocketQA: An Optimized Training Approach to Dense Passage Retrieval for Open-Domain Question Answering
[2]: Conditional Response Augmentation for Dialogue using Knowledge Distillation

**Paper Topic And Main Contributions:**

This paper studies the false negative problems of conversational response selection tasks. For the purpose of better negative sampling, this paper proposes a variational approach that can consider various characteristics (or, facets) and a novel framework TRIGGER to dynamically update negatives according to the current model capacity. The proposed approach is applied in multiple retrieval models and evaluated on two famous response selection benchmarks.

**Questions For The Authors:**

1. Can the proposed method further improve performance with better PLMs?
2. How expensive the proposed method in the perspective of computational costs in comparison with the competitors?

**Reasons To Accept:**

1. The paper is generally well written and easy to follow.
2. The tackled problem (false negatives in negative sampling) is important and the proposed method is reasonably designed to address the problem.
3. The proposed method achieves the new state-of-the-art performances in Ubuntu and Douban datasets.
4. The pilot study (Section 2) that manually inspects the ratio of false negatives in various methods is meaningful and would inspire a bunch of future researches.

**Reasons To Reject:**

There is no serious reason to reject this paper. But, I have a few concerns regarding the novelty of the proposed method and experiments. It would be helpful to strengthen the following points:
1. The distinction from CVAE (conditional variational auto-encoder): Though this paper tackles its own problem in response selection tasks and proposes a novel method, the proposed method seems sharing a lot of aspects with CVAE architecture in my perspective. Highlighting the novelty of leveraging ''conversational'' aspects would be a good initial point for this concern.
2. Weak baselines: Among baselines, I realized that recent knowledge distillation-based approaches methods are not considered, such as [1] in passage retrieval and [2] in response selection tasks. Though they are not seriously relevant and important to cite, comparing the proposed method with them would make the contribution of this paper much clearer.
3. Marginal improvements over baselines: Though the proposed method achieves the largest improvements among the competitors, I wonder if we should pursue such a sophisticated framework to achieve this level of performance. I think further improvements would be achievable with other PLMs better than BERT, such as RoBERTa and T5 (or, more recently vicuna or llama-2). It would be greatly helpful to discuss which aspects of the proposed method and experiments are meaningful in the era of LLMs, such as computational costs or real-world scenarios.

**Reproducibility:**

4: Could mostly reproduce the results, but there may be some variation because of sample variance or minor variations in their interpretation of the protocol or method.

**Reviewer Confidence:**

4: Quite sure. I tried to check the important points carefully. It's unlikely, though conceivable, that I missed something that should affect my ratings.

---

> ### Author Rebuttal · Authors · 2023-08-29
>
> Thanks for your time and expertise invested in reviewing our submission! We are grateful to receive your appraisal that our work provides a well-designed method to tackle the problem and would inspire a bunch of future research. This is indeed our original aspiration. Below, we will address each point individually.
>
> **Q1: The distinction from CVAE.**
>
> **A1**: Good question! Our proposed Sequential Variational Ladder Auto-Encoder (SVLAE) shares similarities with the CVAE in that they both involve one or more latent labels. However, SVLAE differs from CVAE in the following ways:
>
> + Previous works using CVAE for dialogue generation [1][2] tend to view the entire dialogue context as a single observed data point $x$, and golden response as the label $y$, seldom considering the internal transition logic between each turn in $x$. In contrast, we treat the dialogue session $u_{1}, u_2, \ldots, u_N, u_{N+1}$ as a sequence of observed data and use both latent labels $y_{1:N}$ and latent feature $z_{1:N}$ to model the sequential transition logic within dialogue.
>
> + Multiple orthogonal characteristics influence dialogue flow, such as dialogue topic, discourse coherence, and speaker emotion. It is challenging to encode these abundant pieces of information with a single discrete or continuous latent variable. To address this, we propose a ladder-like architecture (Figure 1) to encode multiple levels of semantic abstraction with $y^{1:L}$ and $z^{1:L}$ in a compositional and disentangled manner, which is a major novelty of our SVLAE.
>
> We are sorry for the ambiguity and will explain it more clearly in our final version.
>
>
> [1]Learning Discourse-level Diversity for Neural Dialog Models using Conditional Variational Autoencoders, ACL 2017.
>
> [2]A Discrete CVAE for Response Generation on Short-Text Conversation, EMNLP 2019.
>
>
>
> **Q2: Knowledge distillation-based approaches.**
>
> **A2**: Thanks for your suggestions! Following your advice, we compare with RocketQA[1] and CRA[2] using BERT and SA-BERT as base retrieval models and the results are shown below:
>
> | | R$_{10}$@1 | R$_{10}$@2 | R$_{10}$@5 |
> |---|---|---|---|
> |BERT| 80.8 | 89.7 | 97.5 |
> |BERT+RocketQA| 81.5 | 90.0 | 97.5 |
> |BERT+CRA| 81.3 | 89.9 | 97.5|
> |BERT+TRIGGER| 81.9 | 90.4 | 97.5 |
>
>
>
> | | R$_{10}$@1 | R$_{10}$@2 | R$_{10}$@5 |
> |---|---|---|---|
> |SA-BERT| 85.5 | 92.8 | 98.3 |
> |SA-BERT+RocketQA| 85.9 | 93.3 | 98.5 |
> |SA-BERT+CRA| 86.0 | 93.2 | 98.3 |
> |SA-BERT+TRIGGER| 87.1 | 93.8 | 98.7 |
>
> The table demonstrates that our method surpasses both CRA and RocketQA, potentially due to our explicit consideration of multiple characteristics within the conversational flow, rendering our approach more adept at uncovering false negatives in open-domain dialogue. In our final version, we will integrate these experiments and reference knowledge distillation-based techniques [1][2].
>
>
>
> [1]RocketQA: An Optimized Training Approach to Dense Passage Retrieval for Open-Domain Question Answering, NAACL 2021.
>
> [2] Conditional Response Augmentation for Dialogue Using Knowledge Distillation. INTERSPEECH 2020.
>
> **Q3: Marginal improvements over baselines.**
>
> **A3**:  We recognize that our approach may not yield a "sharp" enhancement in the retrieval performance of the Douban or Ubuntu benchmark, given their inherent dataset complexity and the limitations of the base retrieval model. Nonetheless, it is crucial to note that our proposed TRIGGER framework significantly bolsters the robustness of retrieval models, as evidenced by Table 6, which is vital for deployment and service. Additionally, our method is compatible with any base retrieval model and can be integrated with more advanced PLM encoders to further augment performance (kindly refer to our response to Q6).
>
> **Q4: Which aspects of the proposed method and experiments are meaningful in the era of LLMs.**
>
> **A4**: Good question! Despite the remarkable performance of LLM, retrieval-based dialogue systems and our proposed negative sampling techniques maintain their distinct advantages:
>
> + Enhanced safety and control. As responses are chosen from a predefined candidate pool, retrieval-based dialogue systems offer greater content control, thereby mitigating the presence of toxic words or hallucinations in responses.
>
> + Reduced computational cost. While LLMs generate high-quality output, the substantial computational expense of APIs (e.g., chatGPT and GPT-4) should not be disregarded. Prior research [1] examines the economic trade-off in deploying dialogue systems for customer service, revealing that LLM (GPT-3) may not be the optimal choice when considering both generation quality and API cost compared to smaller models.
>
> + Retrieval-augmented LLM. Many researchers contend that retrieval should play a vital role in next-generation language models to enhance their factuality, interpretability, and trustworthiness. For example, LLMs are less likely to hallucinate when provided with retrieved knowledge from a given knowledge base [2][3][4]. Conversely, LLM output can be unstable due to sensitivity to prompts and decoding strategies. A potential solution is to first generate multiple candidates by using LLM to simulate various personas [5] or engage in multi-agent debates [6]. A retrieval module may then select the most suitable candidate as a response.
>
> + Tool retrieval for LLM. External tools or APIs remain necessary for addressing higher-level tasks with LLMs. In this context, a retrieval system is required to select an appropriate tool (API) from thousands of API documents based on specific human instructions [7]. Given the functional similarity of APIs and the existence of multiple solution paths (chains of API calls), randomly sampled negative APIs may suffer from false negative issues, and our approach offers a potential remedy for this problem.
>
>
> [1]The economic trade-offs of large language models: A case study, ACL 2023.
>
> [2]Check Your Facts and Try Again: Improving Large Language Models
> with External Knowledge and Automated Feedback, Arxiv 2023.
>
> [3]Enabling large language models to generate text with citations, Arxiv 2023
>
> [4]Rarr: Researching and revising what language models say, using language models.
>
> [5]Unleashing cognitive synergy in large language models: A task-solving agent through multi-persona self-collaboration, Arxiv 2023
>
> [6]Improving factuality and reasoning in language models through multiagent debate, Arxiv 2023.
>
> [7]ToolLLM: Facilitating Large Language Models to Master 16000+ Real-world APIs, Arxiv 2023.
>
>
>
> **Q5: I think further improvements would be achievable with other PLMs better than BERT, such as RoBERTa and T5 (or, more recently vicuna or llama-2). Can the proposed method further improve performance with better PLMs?**
>
> **A5**: Absolutely. Although our experiments primarily employ BERT-based models to maintain consistency with prior literature [1][2], our proposed TRIGGER framework remains agnostic and orthogonal to PLM architecture. This framework can be effortlessly integrated with any existing PLM to bolster its performance in dialogue retrieval. For instance, when combined with DeBERTa [3], a more recent and potent backbone, dialogue retrieval performance could be further enhanced, as illustrated below:
>
> | | R$_{10}$@1 | R$_{10}$@2 | R$_{10}$@5 |
> |---|----|----|----|
> |DeBERTa |  85.4 |  92.6 |  98.2 |
> |DeBERTa+TRIGGER |  86.5 | 93.2 | 98.3 |
> |DeBERTa-FP | 92.0 | 96.6 | 99.5 |
> |DeBERTa-FP+TRIGGER | 92.8 | 97.2 | 99.6 |
>
>
> We believe that these results provide valuable insights into the effectiveness and versatility of our approach across different model backbones.
>
>
> [1]Dialogue Response Selection with Hierarchical Curriculum Learning, ACL 2021.
>
> [2]Synthesizing Adversarial Negative Responses for Robust Response Ranking and Evaluation, ACL 2021.
>
> [3]DEBERTA: DECODING-ENHANCED BERT WITH DISENTANGLED ATTENTION, ICLR 2021
>
>
>
>
> **Q6: The computational costs of the proposed method?**
>
> **A6**: We appreciate the reviewer's apprehension. Our proposed method concentrates on negative sampling, offering more beneficial responses and co-evolving with the retrieval model. During inference, only the retrieval model D requires deployment, ensuring that our approach imposes no supplementary computational costs or latency during inference.
>
>
>
> **Q7: Further definition of "model capacity" would improve the paper reading.**
>
> **A7**: Thanks for your suggestion! Since we mainly focus on multi-turn response selection task in this study, the "model capacity" in our manuscript refers to the ability of a retrieval model $D(\cdot \mid c,\mathcal{R})$ to discern multiple candidates in $\mathcal{R}$ and find the appropriate response $r^+$ without fooled by the negatives. We will definitely follow your advice and add the detailed explanation in Section 4 in the final version to enhance the paper's readability.
>
>
> **Q8: It is hard to understand how can I interpret the results in Section 6.4. Do the results contribute to the performance improvements in the above sections?**
>
> **A8**: Good question! The impetus for Section 6.4 is to offer a proof-of-existence and an intuitive comprehension of the transition logic for our readers. Specifically, we can observe certain transition patterns in the Ubuntu Corpus from Table 7 and Table 8. For instance, the most prevalent pattern is $233\to155$, signifying that a concise instruction on paste operations in the pastebin.ubuntu website (latent label $233$) typically precedes a reference to a URL (latent label $155$) within a conversation session.
> In summary, this subsection affirms the efficacy of the proposed Sequential Variational Ladder Auto-Encoder (SVLAE) in capturing transition patterns within a conversation corpus in a more intuitive and qualitative manner, thereby facilitating an understanding of our method's effectiveness.
>
> **Q9: Grammar typos and presentation improvement**
>
> **A9**: Thanks for your suggestions! we will definitely revise the grammar typos and incorporate your advice on presentation into our final version.

---

### Meta-Review · Area_Chair_1hW9 · 2023-09-08

**Recommendation:** 3

**Metareview:**

This paper delves into the false negative issues within conversational response selection tasks. To enhance negative sampling, the authors put forth a variational method accounting for diverse characteristics, along with the TRIGGER framework to dynamically refresh negatives based on the model's evolving capacity. This approach is tested on various retrieval models and validated using two renowned benchmarks: the Ubuntu and Douban datasets.

Soundness scores stand at (4, 3, 4), reflecting reviewers' consensus on the paper's solidity. The presented claims are well supported by the studies and experimental data.

On the other hand, excitement scores read (3, 2, 3). A primary critique highlights the proposed solution as overly complex, with limited generalizability and practical application. The performance improvement is marginal. However, there is still a notable highlight is section 2, which presents an insightful human-centric pilot study on false negatives.

---

### Decision · Program_Chairs · 2023-10-07

**Decision:**

Accept-Findings

**Comment:**

This paper delves into the false negative issues within conversational response selection tasks. To enhance negative sampling, the authors put forth a variational method accounting for diverse characteristics, along with the TRIGGER framework to dynamically refresh negatives based on the model's evolving capacity. This approach is tested on various retrieval models and validated using two renowned benchmarks: the Ubuntu and Douban datasets.

Soundness scores stand at (4, 3, 4), reflecting reviewers' consensus on the paper's solidity. The presented claims are well supported by the studies and experimental data.

On the other hand, excitement scores read (3, 2, 3). A primary critique highlights the proposed solution as overly complex, with limited generalizability and practical application. The performance improvement is marginal. However, there is still a notable highlight is section 2, which presents an insightful human-centric pilot study on false negatives.